# Effects of Thymol Supplementation on Goat Rumen Fermentation and Rumen Microbiota In Vitro

**DOI:** 10.3390/microorganisms8081160

**Published:** 2020-07-30

**Authors:** Jiangkun Yu, Liyuan Cai, Jiacai Zhang, Ao Yang, Yanan Wang, Lei Zhang, Le Luo Guan, Desheng Qi

**Affiliations:** 1Department of Animal Nutrition and Feed Science, Huazhong Agricultural University, Wuhan 430070, China; YJK555@webmail.hzau.edu.cn (J.Y.); doriacai@mail.hzau.edu.cn (L.C.); zjc404@webmail.hzau.edu.cn (J.Z.); yangao@webmail.hzau.edu.cn (A.Y.); YNWang@webmail.hzau.edu.cn (Y.W.); zhanglei6@webmail.hzau.edu.cn (L.Z.); 2Department of Agricultural, Food and Nutritional Science, University of Alberta, Edmonton, AB T6G 2P5, Canada; lguan@ualberta.ca

**Keywords:** thymol, rumen fermentation, rumen microbiota, goats

## Abstract

This study was performed to explore the predominant responses of rumen microbiota with thymol supplementation as well as effective dose of thymol on rumen fermentation. Thymol at different concentrations, i.e., 0, 100 mg/L, 200 mg/L, and 400 mg/L (four groups × five replications) was applied for 24 h of fermentation in a rumen fluid incubation system. Illumina MiSeq sequencing was applied to investigate the ruminal microbes in addition to the examination of rumen fermentation. Thymol doses reached 200 mg/L and significantly decreased (*p* < 0.05) total gas production (TGP) and methane production; the production of total volatile fatty acids (VFA), propionate, and ammonia nitrogen, and the digestibility of dry matter and organic matter were apparently decreased (*p* < 0.05) when the thymol dose reached 400 mg/L. A thymol dose of 200 mg/L significantly affected (*p* < 0.05) the relative abundance of 14 genera of bacteria, three species of archaea, and two genera of protozoa. Network analysis showed that bacteria, archaea, and protozoa significantly correlated with methane production and VFA production. This study indicates an optimal dose of thymol at 200 mg/L to facilitate rumen fermentation, the critical roles of bacteria in rumen fermentation, and their interactions with the archaea and protozoa.

## 1. Introduction

Ruminants are unique livestock species for their exquisite rumen structure that harbors a complex microbiota responsible for converting indigestible plant mass into energy available to the host [1]. The rumen microbial population includes members belonging to all three domains, i.e., bacteria, archaea, and eukarya (ciliate protozoa and anaerobic fungi) [2] consisting of an enormous quantities of cellulolytic, amylolytic, proteolytic and lipolytic microorganisms. Rumen microbial fermentation products, mainly volatile fatty acids (VFA), contribute to 70% of the daily energy requirement [3,4]. However, as a result of rumen fermentation, ruminants also contribute greatly to the production of the greenhouse gas methane (CH_4_) [5,6]. Therefore, manipulation of rumen microbial fermentation could both improve animal production and alleviate adverse environment impacts.

To modify rumen fermentation, feed additives are commonly applied. Among the feed additives, essential oils (EOs) are considered promising with their strong antiseptic and antimicrobial features [7]. EOs have also been reported to benefit the performance of swine and poultry [8]. In ruminants, supplementation with EOs positively affected starch and protein degradation, production of ammonia and VFA, and moreover reduced methane emissions [9]. However, EOs possess a highly variable composition attributed to plant species, the growth environment of the plant, stage of plant growth, and part of the plant used for extraction [10]. Thus, the application of EOs as a feed additive is constrained by the absence of production standards (proportion of active compounds) and acknowledged quantified dosages. Therefore, further studies are required to study the active compounds in EOs and their impact on rumen fermentation for future practical applications.

Thymol (5-methyl-2-isopropylphenol) is one of the active compounds in EOs derived from *Thymus* and *Origanum* plants [11,12]. Dosages of thymol higher than 240 mg/L have been reported to affect rumen fermentation with a decrease in methanogenesis and altered rumen bacteria composition in lactating cows in vitro [13]. A similar effect on rumen methane reduction was found by Castañeda-Correa et al. [14] with 200 mg/L thymol supplemented in vitro, whereas no effect was found on dry matter digestibility and methanogenic bacteria except for individual bacterial population changes within rumen microorganisms. Experiments in vivo have been carried out in sheep by Biricik et al. [15], showing that thymol up to 300 mg/kg in the diet did not affect dry matter intake or feed conversion, although the total VFA concentration was increased. Blends of EOs or active compounds containing thymol have been studied in buffalo [16], dairy cow [17], sheep [18], and beef steers [19]; however, the results were not consistent in terms of methane production or rumen fermentation. So far, the effective dose of thymol on rumen fermentation and methane production remains inconclusive. Furthermore, how the microbial interactions within bacteria, archaea, and protozoa regulate rumen fermentation production are far from being understood. Hence, the objectives of this study were to (1) explore the effect of different concentrations of thymol on rumen fermentation and rumen microbiota including bacteria, archaea, and protozoa in goats using an in vitro approach; and (2) to explore the possible correlation of rumen fermentation production as well as methanogenesis with regard to alterations in the composition of bacteria, archaea, and protozoa. This study could lay the foundation for a more comprehensive understanding of how the rumen microbiota responds to a shift in the rumen microenvironment.

## 2. Materials and Methods 

### 2.1. Animal and Rumen Fluid Collection

Animal procedures were performed according to the Guidelines of the Institutional Animal Care and Use Committees at the Huazhong Agricultural University (Protocol No. HZAUMO-2016-054). Three Crossbreed Boer female goats (37.34 ± 2.16 kg) were used as rumen fluid donors in this study. Vaccination and other prophylactic measures described by Vatta et al. [20] were carried out before the experiment to ensure that the goats were healthy. Goats were fed twice daily (08:00 h and 17:00 h); an excess amount of basal diet (1.2 times of feed intake of goats in the pilot test) and free access to water were provided to goats 14 d before of rumen fluid sampling. The basal diet with a total mixed ration (TMR) consisted of (DM basis) alfalfa (560 g/kg ), group corn (270 g/kg), and a concentrate mixture (170 g/kg) (dry matter 86.94%, crude protein 13.44%, ether extract 1.83%, crude fiber 17.09%, nitrogen free extract 41.17%, ash 13.41%, and gross energy 15.13 MJ/kg). Rumen fluid was collected 3 h after the morning feed using a stomach tube. The rumen content was strained through four layers of sterilised cheesecloth before inoculation for fermentation. 

### 2.2. In Vitro Batch Fermentation

A commercial product of thymol (Shanghai Yuanye Biotechnology Co., Ltd., Shanghai, China) with a purity of ≥ 98% was applied in the present study. Four different doses of thymol (0, control group (CON); 100 mg/L, low concentration group (LC); 200 mg/L, medium concentration group (MC); and 400 mg/L, high concentration group (HC) with 5 replications in each group were tested in vitro during fermentation. The incubation was carried out with 30 mL of medium (10 mL rumen fluid and 20 mL double strength buffer) in a 100 mL calibrated glass syringe following the procedure described by Menke and Steingass [21]. The incubation was maintained at 39 °C for 24 h, and the following parameters were measured after incubation: total gas production (TGP), methane production, ammonia nitrogen (NH_3_-N), VFA including acetate, propionate, butyrate, isobutyrate, valerate, and isovalerate, total VFA, the ratio of acetate to propionate, in vitro dry matter digestibility (IVDMD), and in vitro organic matter digestibility (IVOMD), as well as the diversity and composition of the rumen microbiota. 

### 2.3. Parameters Analysis Post Rumen Fermentation

Total gas production (TGP) was recorded from the calibrated scale on the syringe, and 5 mL of gas was subsampled from the headspace of syringe for CH_4_ concentrations using gas chromatography (GC) (Agilent 7890A, Agilent Technologies Inc., Santa Clara, CA, USA) with a flame ionization detector (FID). The gas flow rates for nitrogen, hydrogen, and air were 40, 40, and 400 mL/min, respectively. The temperatures of the injector oven, column oven, and detector were 55, 55, and 250 °C, respectively.

The supernatants were used for NH_3_-N and VFA analysis. The NH_3_-N concentration was determined following the phenolhypochlorite method [22]. Concentrations of individual VFA were determined using gas chromatography (GC) followed the method described by Yang et al. [23]. Briefly, 0.2 mL of supernatant was added to 1 mL of meta-phosphate acid 25% (*w*/*v*) and centrifuged at 10,000 r/min for 10 min. The sample was then injected into Chrompack CP-Wax 52 CB fused silica column (30 m × 0.53 mm i.d., 1.0 mm film thickness) on GC (Model 2010, Shimazu, Japan) equipped with FID (Flame Ionization Detector). 

For IVDMD and IVOMD determination, the contents were digested with neutral detergent solution and the undigested feed was recovered on crucibles, and washed and dried. The IVDMD and IVOMD of substrate were calculated by subtracting the value of dry matter (DM) and organic matter (OM) from the DM and OM incubated in the syringe, respectively [24].

### 2.4. Analysis of Rumen Microbial Communities

An aliquot of rumen fluid (1 mL) was taken for microbial profile analysis and stored at −80 °C until analysis. Total DNA was extracted following the procedure described by Yu and Morrison [25]. The quantity and quality of extracted DNAs were measured using a NanoDrop ND-1000 spectrophotometer (Thermo Fisher Scientific, Waltham, MA, USA) and agarose gel electrophoresis, respectively. Primers [26,27] (Table 1) with special barcodes were applied for PCR amplification of bacterial V3 to V4 regions of 16S rRNA genes, archaeal V6 to V8 regions of 16S rRNA genes [28], and protozoan V5 to V8 regions of 18S rRNA genes [29] using the following programs. For bacteria, 98 °C for 2 min; 25 of cycles, 98 °C for 15 s, 55 °C for 30 s, 72 °C for 30 s, and then 72 °C for 5 min; for archaea, 98 °C for 2 min; 30 of cycles, 98 °C for 15 s, 55 °C for 30 s, 72 °C for 30 s, and then 72 °C for 5 min; and for protozoa, 95 °C for 3 min; 35 of cycles, 95 °C for 30 s, 55 °C for 30 s, 72 °C for 45 s, and then 72 °C for 10 min. Respective PCR amplicons were purified using Agencourt AMPure Beads (Beckman Coulter, Indianapolis, IN) and the concentrations were quantified using the PicoGreen dsDNA Assay Kit (Invitrogen, Carlsbad, CA, USA). Then, amplicons were pooled in equal amounts; bacterial amplicons were subjected to pair-end 2 × 250, and archaeal and protozoan amplicons were subjected to pair-end 2 × 300 bp sequencing using the Illlumina MiSeq platform at Shanghai Personal Biotechnology Co., Ltd. (Shanghai, China) and Shanghai Majorbio Bio-pharm Technology Co., Ltd (Shanghai, China). 

Sequencing reads were pre-processed using QIIME2 (ver. 2018.8) and its plugins [30]. Specifically, the “demux” plugin (https://github.com/qiime2/q2-demux) was used to import the demultiplexed paired-end sequencing reads and to create the “artifact” file (i.e., QIIME2 data format required for subsequent analyses). The “dada2” pipeline [31] was used for quality filtering and chimera removing to trim primers, to truncate forward and reverse reads, and to assembly reads into amplicon sequence variants (ASVs). Only ASV with relative abundance ≥ 0.05% was included for taxonomic analysis. Taxonomy was assigned using a naïve Bayes classifier [32] trained on the Silva_132_99% (Silva 132) database [33] for bacteria, and the Rumen and Intestinal Methanogen database (RIM-DB [28]) for archaea. Ciliate protozoa taxonomy was assigned using naïve Bayes classifier based on sequences data (168 almost full-length (≥ 1500 bp) intestinal ciliate 18S rRNA gene sequences) summarized by Kittelmann et al. [29]. The taxonomic lineage for each ASV was identified and summarized at phylum, class, order, family, genus, and species levels. A rooted phylogenetic tree was created through de novo multiple sequence alignment using MAFFT [34], positional conservation and gap filtering was performed using MASK [35], and FastTree-2 was applied to generate a tree from the masked alignment [36]. Alpha diversity was estimated using the Chao1 index for community richness, the Pielou index for community evenness, and the Shannon index for community diversity. Beta diversity analysis was performed using the distance matrix generated from the weighted UniFrac phylogenetic metric [37] and was plotted in a principle coordinate analysis (PCoA) using the R (ver. 3.5.2) [38].

### 2.5. Network Analysis

Correlation networks based on Spearman correlation analysis were generated to explore the relationships among rumen microbiota and rumen fermentation products. To generate the Spearman correlation matrix, both rumen microbial communities and rumen fermentation parameters were collected for correlation analysis with Hmisc (ver. 4.4-0) [39] in R. In brief, rumen microbial communities consisted of bacterial and protozoan communities with relative abundances at genus level, and archaeal communities with relative abundances at species level. Methane production based on organic matter digested, individual, and total VFA productions, as well as A/P ratio with the determined values, were used as representatives of rumen fermentation characteristics. Only bacterial and protozoa genera and archaeal species with a relative abundance higher than 0.5% were selected at relevant levels. The co-occurrence network was built based on correlation coefficients and *p*-values. All the *p*-values were adjusted using the Benjamini and Hochberg false discovery rate (FDR) controlling procedure [40]. The cutoff of correlation coefficients and FDR-adjusted *p*-values were determined as 0.75 and 0.001, respectively. The edges and nodes generated from selected conditions were subsequently taken as input files in Cytoscape (ver. 3.7.2) [41] to generate the final network image. The layout of the final network was attributed to circle according to degree (number of adjacent edges) of related nodes. 

### 2.6. Statistical Analysis

All the fermentation parameters, including total gas and CH_4_ production, ruminal pH, VFA, NH_3_, IVDMD, and IVOMD were analysed in R (ver. 3.5.2) with Kruskal–Wallis test and Dunn test for multiple pairwise-comparison between groups. Similarly, the microbial composition at different levels as well as indices of alpha diversity were analyzed using Kruskal–Wallis test and Dunn test. Beta diversity was calculated using weighted UniFrac distance, and a permutational multivariate analysis of variance (PERMANOVA) [42] was performed in QIIME2 to confirm significant differences between groups. For all the statistical tests, *p*-value of each test was adjusted into false discovery rate (FDR) using the Benjamini–Hochberg algorithm [40], the threshold of FDR-adjusted *p*-values was set to 0.05 to determine the significance, and a trend was recognized when 0.10 > *p* ≥ 0.05. 

### 2.7. Data Submission

All the amplicon sequencing dataset in this study were submitted to NCBI Sequence Read Archive (SRA) under the accession number PRJNA639952.

## 3. Results

### 3.1. Total Gas and Methane Production, pH, Digestibility, and Ammonia Concentration

Compared with control (CON) group, a significant increase in ruminal pH (*p* < 0.01) as well as a notable reduction (*p* < 0.01) in TGP was observed in the medium concentration (MC) and high concentration (HC) groups supplemented with thymol (Table 2). CH_4_ production based on fermented dry matter (DM) or organic matter (OM) was significantly decreased (*p* < 0.01) in the MC and HC groups (Table 2). Both IVDMD and IVOMD were only significantly decreased (*p* < 0.05) in the HC group with a considerable decline of over 10% (Table 2). Thymol at a dose of 200 mg/L and 400 mg/L apparently reduced (*p* < 0.01) rumen NH_3_-N concentration compared with the CON group (Table 2). 

### 3.2. Volatile Fatty Acids (VFA) Concentrations

Total VFA concentration was reduced by highest dose of thymol (*p* < 0.05) and did not differ (*p* > 0.1) between the LC or MC group and CON group (Table 2). The acetate concentration did not show significant difference (*p* > 0.05) in thymol supplemented groups compared with the CON group. Intriguingly, the acetate concentration was significantly lower in the HC group (*p* < 0.05) compared with the LC group. Supplementation with thymol exerted a similar impact on the concentrations of propionate, and a tendency to increase was shown in the LC and MC groups while a significant decrease (*p* < 0.05) was observed in the HC group compared with the LC group (Table 2). The butyrate concentration in the HC group were significantly decreased (*p* < 0.05) compared with the LC or MC group. The acetate/propionate (A/P) ratio was higher (*p* < 0.05) only in the HC group compared with the CON group (Table 2). The isobutyrate concentration was dramatically lower (*p* < 0.01) in the MC, and HC groups compared with the CON group, and did not differ (*p* > 0.05) between thymol supplemented groups (Table 2). The valerate concentration decreased with the increasing doses of thymol and reached significant level in the MC and HC groups (*p* < 0.01) compared with CON group (Table 2). Only a trend (*p* = 0.080) was observed for the isovalerate concentration with thymol supplementation (Table 2).

### 3.3. Rumen Bacteria

From 20 samples, a total of 328,462 high quality sequences were obtained following the removal of low-quality or chimeric sequences, with an average of 16,423 ± 3888 sequence counts per sample. The composition of bacteria across the 20 samples was dominated by the phyla Bacteroidetes, Firmicutes, Synergistetes, Proteobacteria, Kiritimatiellaeota, Actinobacteria, and Spirochaetes (Figure 1; Appendix A). Notably, the phyla Bacteroidetes and Firmicutes accounted for over 70% of all bacteria across these samples. With an increasing dose of thymol, the relative abundance of Bacteroidetes significantly declined (*p* < 0.01), while a gradient increase in relative abundance was observed for Firmicutes (*p* < 0.01). At the genus level, 14 dominant genera were observed across the samples (Figure 1; Appendix A). Among them, *Prevotella* 1, the Rikenellaceae RC9 gut group, *Succiniclasticum*, *Streptococcus*, *Pseudobutyrivibrio*, and *Fretibacterium* were the top six genera. In addition, dramatic changes appeared in *Prevotella* 1, *Streptococcus*, and *Pseudobutyrivibrio*. *Prevotella* 1 possessed a predominant relative abundance of over 25% in the CON group, but nearly disappeared in the HC group with a relative abundance lower than 1%. On the contrary, the relative abundances of *Succiniclasticum*, *Streptococcus*, and *Pseudobutyrivibrio* were significantly increased with an increasing dose of thymol.

The alpha diversity was estimated by the Chao1, Pielou, and Shannon indices for bacterial richness, evenness, and diversity, respectively (Figure 2; Appendix A). Compared with the CON group, all three indices were affected by thymol addition. The Shannon index decreased linearly (*p* < 0.01) with an increased dose of thymol supplementation. The Chao1 and Pielou indices also decreased (*p* < 0.01) with thymol supplementation but did not differ between the LC and MC groups. The beta diversity of rumen bacteria was shown by weighted UniFrac PCoA (Figure 3a). Groups treated with different thymol doses were almost completely separated with four clusters. The significant differences between groups were confirmed by the PERMANOVA analysis, which showed a significant (*p* = 0.001) effect of thymol treatments on weighted UniFrac distances.

### 3.4. Rumen Archaea

A total 462,270 sequences across all the samples were obtained after low-quality or chimeric sequences were filtered, with an average of 23,114 ± 2652 sequence counts per sample. More than 99.9% of archaea taxa were aligned to the phylum Euryarchaeota, and all the classes belong to this phylum, i.e., Methanobacteria, unclassified Euryarchaeota, Methanomicrobia, Thermoplasmata, and unclassified Archaea with the relative abundance decreasing across all the samples (Figure 4; Appendix A). Methanobacteria as the most dominant class comprised a proportion of over 83% across all the samples. Supplementation with thymol exerted little significant impact on archaea composition at the class level, except for the tendency to decrease the relative abundance of Methanomicrobia (*p* = 0.080). At the species level, the *Methanobrevibacter gottschalkii* clade was present with the most dominant relative abundance of over 73%, followed by unclassified Euryarchaeota and the *Methanobrevibacter boviskoreani* clade, unclassified Methanomicrobia, *Methanosphaera* sp. Group5, unclassified Methanomassiliicoccaceae, and *Methanobrevibacter ruminantium* clade (Figure 4; Appendix A). The higher relative abundance in the LC and MC groups as well as the lower relative abundance in the HC group compared with the CON group led to significant differences (*p* < 0.05) in the *Methanobrevibacter boviskoreani* clade and *Methanobrevibacter ruminantium* clade when comparing the HC group with the LC and MC groups. An increasing dose of thymol linearly reduced the abundance of unclassified Methanomassiliicoccaceae (*p* = 0.008). A tendency (*p* = 058) to decrease was observed in unclassified Methanomicrobia, while no difference was found in the *Methanobrevibacter gottschalkii* clade and unclassified Euryarchaeota between the control and thymol supplemented groups.

The alpha diversity of rumen archaea was estimated using the Chao1, Pielou, and Shannon indices. The Chao1 and Pielou indices were significantly affected (*p* < 0.05) by thymol addition, while no difference (*p* > 0.05) was observed in the Shannon index (Figure 2; Appendix A). Only a high dose of thymol significantly increased the Pielou index, and the Chao1 index in the HC group was significantly lower than in the LC and MC groups. The beta diversity was displayed by weighted UniFrac PCoA (Figure 3b). Only the community of the HC group was obviously distinct from the other three groups, and a significant effect (*p* = 0.011) of thymol supplementation was determined on weighted UniFrac distances.

### 3.5. Rumen Ciliate Protozoa

A total 552,731 sequences across all the samples were obtained after low-quality or chimeric sequences were filtered, with an average of 27,636.55 ± 12,485.84 sequence counts per sample. All the ciliate protozoa were aligned to class Litostomatea within phylum Ciliophora, and the most predominant order was Entodiniomorphida with a relative abundance higher than 93%, followed by order unclassified Litostomatea with a relative abundance around 6%. Four families were identified, i.e., Ophryoscolecidae, unclassified Litostomatea, Isotrichidae, and unclassified Entodiniomorphida from highly abundant to less abundant across all the samples (Figure 5; Appendix A). The only significant difference (*p* < 0.05) between the HC and LC groups was detected for Isotrichidae at the family level (Figure 5). At the genus level, the five most abundant genera, in total accounting for over 93% in relative abundance, were *Entodinium*, unclassified *Litostomatea*, *Enoploplastron*, *Isotricha*, and *Diploplastron-Eremoplastron* (Figure 5). Similar impacts were driven by the addition of thymol on *Polyplastron* and *Diploplastron-Eremoplastron* with higher relative abundance in the LC group and lower relative abundance in the MC and HC groups, which indicated a significant difference in HC group compared with the LC group. A trend (*p* = 0.060) was observed in relative abundance of *Isotricha*. The unclassified Litostomatea did not differ (*p* > 0.05) for all the treatment groups compared with the control group.

The alpha diversity of rumen ciliate protozoa was estimated by the Chao1, Pielou, and Shannon indices (Figure 2; Appendix A). Similar fluctuations were observed in the Pielou and Shannon indices, which increased in LC and decreased in MC and HC, resulting a significant difference (*p* < 0.05) between the low dose and high dose addition groups. For the Chao1 index, a significant difference (*p* < 0.05) was found between the LC and HC groups. The beta diversity displayed by weighted UniFrac PCoA and indicated no differences between thymol treatment groups and the CON group (Figure 3c). No distinct separation between groups was found and no significant effect (*p* = 0.142) was detected by PERMANOVA on weighted UniFrac distances for the thymol treatments.

### 3.6. Co-Occurrence Network Analysis

The resulting co-occurrence network (Figure 6 and Appendix A) comprised of 41 nodes and 136 edges. Interactions between microbial organisms including bacteria, archaea, and ciliate protozoa, as well as phenotype traits involved with methane production and VFA production were revealed in the relevance network. The bacterial genera *Prevotella* 1, *Quinella*, Veillonellaceae UCG-001, and uncultured rumen bacterium2, and the archaeal species unclassified Methanomassiliicoccaceae were commonly positively correlated with the production of methane, valerate, and isobutyrate. Additionally, two bacterial genera, i.e., *Streptococcus* and *Pseudobutyrivibrio*, were negatively correlated with methane and valerate production, and a negative correlation was found between *Pseudobutyrivibrio* and isobutyrate. The production of methane and valerate were also found to be positively correlated with unclassified Bacteria, the Ruminococcaceae NK4A214 group, Prevotellaceae UCG-003, and uncultured rumen bacterium4. The special bacterial genus *Treponema* 2 served as a bridge to connect two groups of nodes in the network and was positively correlated with *Eubacterium ruminantium* group, *Prevotella* 1, unclassified bacteria, and the protozoan genus *Diploplastron-Eremoplastron*, and strongly negatively correlated with *Streptococcus.* In addition, *Treponema* 2 was positively correlated with propionate production, which also showed a positive correlation with Ruminococcaceae UCG-002, uncultured rumen bacterium3, *Succinivibrio*, and the protozoan genus *Isotricha*. No correlation was obtained between microbial profiles and total VFA production, which positively correlated only with individual VFA production including acetate, propionate, isovalerate, and butyrate. As for the A/P ratio, negative correlations were observed with *Prevotella* 1, *Quinella*, unclassified bacteria, and the archaeal species unclassified Methanomassiliicoccaceae as well as valerate production, and only *Streptococcus* and *Pseudobutyrivibrio* were positively correlated with the A/P ratio. Within the VFA production traits, positive correlations were detected between acetate, propionate, and isovalterate, and isobutyrate displayed a positive correlation with valerate. Within the bacterial genera, *Streptococcus* and *Pseudobutyrivibrio* were negatively correlated with the majority of adjacent bacterial genera. In contrast, *Prevotella* 1, *Quinella*, Veillonellaceae UCG-001, and unclassified Prevotellaceae were positively correlated with most of the associated genera. Positive correlations were obtained between these two groups of organisms for sharing similar correlations. *Ruminobacter* was observed to be positively correlated with uncultured rumen bacterium1. Within the archaeal species, only the *Methanobrevibacter gottschalkii* clade was positively correlated with unclassified Euryarchaeota. No correlations were discovered within the two protozoan genera observed in this network.

## 4. Discussion

### 4.1. Rumen Fermentation Production

Decreased total gas production as well as increased ruminal pH were observed when the dose of thymol reached 200 mg/L in the present study; this result is consistent with studies by Benchaar et al. [43] and Kamalak et al. [44]. The inhibition of total gas production, especially in the HC group, mainly resulted from the overall suppression of the activities of rumen microbes during fermentation. Although the pH value did increase from 6.1 to 6.3 after thymol addition, this may have had little impact on rumen fermentation due to the minor rise in view of the wide range of normal ruminal from 5.5 to 7.0 [45]. The effective dose of thymol on methanogenesis inhibition was higher than 200 mg/L in studies performed by Evans and Martin [12] and Macheboeuf et al. [46]; consistent with this, the present study observed a significant decrease in methane production with 200 mg/L and 400 mg/L of thymol supplementation. In the current study, the digestibility of DM and OM was greatly decreased by 400 mg/L thymol addition, which might indicate an inhibitory effect of a high dose of thymol on rumen nutrient digestion; this adverse effect was reported earlier by Cobellis et al. [9]. Nevertheless, Vendramini et al. [47] found no effect of an EO blend containing thymol on rumen DM digestibility when fed to lactating dairy cows, and an increase in DM apparent digestibility was reported by de Souza et al. [48] after feeding heifers with an EO blend. These inconsistent results could be attributed to different conditions such as animal species and the composition of the EO applied in the different studies. The dramatic decline in ammonia N in this study could be attributed to the inhibition of deamination caused by thymol, first reported by Brochers [49]. Additionally, Cobellis et al. [10] suggested an inhibitory effect of thymol on ammonia-producing bacteria in the rumen.

The significant shift in total VFA production caused by 400 mg/L thymol indicates that a high dose of thymol can exert a detrimental effect on rumen fermentation due to the vital role of VFA as a major energy source for the host. Similar to this observation, Castillejos et al. [50] reported an adverse effect of thymol at dose of 500 mg/L on total ruminal VFA production. The significant difference in the acetate production between the LC and HC groups indicates the different effects driven by the low and high doses. Thymol at a low concentration tended to increase acetate production in the rumen while a high dose tended to decrease acetate; a similar decrease tendency (*p* = 0.054) was found when 250 g/kg bioactive compounds of EOs fed to beef cattle [51], and it was seldom reported for the effect of thymol with a low dose of 100 mg/L on ruminal acetate production. Variable results regarding rumen propionate production have been obtained in different studies. Increased propionate proportions were reported after 500 mg/L of thyme oil supplementation to beef cattle in vitro [52] and 5 g/day fed to Holstein calves [53]. Nevertheless, Pirondini et al. [54] and Oh et al. [55] reported a decrease and no effect on propionate production by dairy cows after thyme oil and commercial EO product supplementation in their studies, respectively. In the current study, only the highest dose of thymol significantly decreased rumen propionate, which simultaneously contributed greatly to the increase in the A/P ratio. This result perhaps signifies a lower fermentation efficiency when the high dose of thymol was supplemented, since there is a negative correlation between propionate production and energy loss by methane emission from fermented feeds [45]. In the present study, a significant decrease in isobutyrate, together with a tendency towards a decreased isovalerate concentration by the addition of thymol were consistent with a decline in the NH_3_-N concentration. Thus, the inhibition of deamination caused by thymol was confirmed by another aspect, since isobutyrate and isovalerate are formed from the catabolism of branched-chain amino acids in the rumen [56]. However, minor effects on isobutyrate and isovalerate concentrations caused by EO blends have been reported in different studies [57,58]; although, simultaneously similar decreases in ammonia concentrations were observed. The declined ammonia concentrations in those studies might be attributed to a higher efficiency in consuming ammonia for microbial protein synthesis in the rumen. Butyrate plays a prominent role as a major energy source for epithelial cells [59] and affects rumen nutrition utilisation and function [60]. In the present study, thymol at doses of 100 mg/L and 200 mg/L tended to increase butyrate concentration while a dramatical decrease was observed with 400 mg/L of thymol supplementation. The increase in butyrate concentration is consistent with previous study by Joch et al. [13] with 240 mg/L thymol supplemented to dairy cow in vitro. The dramatical decrease in butyrate concentration caused by highest dose implicates a strong inhibitory effect of high dose thymol in the present study, which is in accord with previous results, i.e., total gas production, IVDMD, and total VFA production. The proportion of valerate in the rumen was reported to be positively correlated with the occurrence of subacute ruminal acidosis [61], and valerate is also thought to affect cellulolytic bacteria in the rumen [62]. An increase in thymol addition linearly decreased the valerate concentration, which may suggest an unfriendly environment for altering the survival of cellulolytic bacteria in the present study. In view of the rumen fermentation parameters affected by thymol supplementation, the appropriate dose of thymol to improve rumen fermentation appears to be 200 mg/L, as dose of thymol as high as 400 mg/L greatly contributed to the inhibition of rumen fermentation.

### 4.2. Rumen Microbiota

Rumen bacteria were dramatically affected by thymol addition, seen from obvious differences in the alpha diversity indices and beta diversity displayed with weighted UniFrac PCoA. Thymol has been reported to have broad antimicrobial activity and act differently against Gram-positive and Gram-negative species [10,11]. Patra and Yu [63] reported an increase in species of the phylum Bacteroidetes and a decrease in Firmicutes in lactating cows using origanum oil containing thymol in vitro. This result was consistent with less susceptibility of Gram-negative bacteria to EOs, attributed to the lipopolysaccharides in its outer membrane [11]. However, in the present study, thymol significantly reduced the abundance of the predominant Gram-negative phylum Bacteroidetes [64] and increased the Gram-positive phylum Firmicutes, exhibiting the better adaption of Gram-positive bacteria. This considerable difference may suggest that pure thymol exerts a stronger effect on Gram-negative bacteria compared with EOs. The genus *Prevotella* 1 is the main group within the genus *Prevotella* and contains the species *Prevotella melaninogenica* and *Prevotella ruminicola* [65]. *P. melaninogenica* is associated with xylan fermentation [66], while *P. ruminicola* is involved in hemicellulose and pectin degradation [67] as well as the metabolism of proteins and peptides [68]. Thus, the decrease in *Prevotella* 1 was probably detrimental to the breakdown of protein and fibrous material, resulting in less methane production since methane production is positively correlated with fibre digestion [69]. Poudel et al. [70] recently noted increases in several species of bacteria belonging to the genus *Prevotella* in calves fed with a blend of EO containing a thymol analogue. To interpret this inconsistency, perhaps a higher resolution of taxonomy is required to explore taxonomic differences at the species level or even lower levels. Members of the Rikenellaceae RC9 gut group are associated with primary or secondary carbohydrate degradation [71]. Consistent with our results, a decrease in the Rikenellaceae RC9 gut group was reported by Oliveira Ramos et al. [72] with tucumã oil supplementation to cows in vitro. The genera *Succiniclasticum*, *Streptococcus*, and *Pseudobutyrivibrio* were apparently increased by a high dose of thymol supplementation. Among these, the genus *Succiniclasticum* comprises species associated with propionate production [73], s species in *Streptococcus* can convert starch to lactate [74], and *Pseudobutyrivibrio* comprises species of butyrate-producing bacteria [75] with strong xylan-degrading potential [76]. Similar increases in the relative abundance of these three genera were observed in dairy cows fed with olive oil [77], in lactating cows using an EO blend in vitro [9], and in steers fed with flax oil [78]. Overall changes in bacterial composition at the genus level indicates an effect on metabolism in the rumen involved in proteolysis, carbohydrate degradation, and organic acid synthesis, with different patterns and to different extents.

Rumen archaea and ciliate protozoa were less affected by thymol supplementation compared with rumen bacteria. The lower susceptibility of archaea compared to bacteria could be attributed to the special cell membrane of archaea stabilised by unique membrane lipids [7]. The more complex structure of eukaryotes could contribute to the better resistance of protozoa. Regarding the alpha diversity, the Chao1 indices of archaea and protozoa were similar with fewer observed bands in the denaturing gradient gel electrophoresis (DGGE) of rumen microbes in cows [79]. Moreover, this study detected a greater richness (higher Chao1 index) of archaea and protozoa with a low dose of thymol. This result implicates that a low dose of thymol supplementation contributes to a better the survival of different species of archaea and protozoa.

Composition of archaea and protozoa were also reported to be affected by thymol or EOs supplementation [10,80]. Within seven of the most predominant species of archaea, similar effects were seen on the *Methanobrevibacter boviskoreani* clade and the *Methanobrevibacter ruminantium* clade with a high dose of thymol supplementation. The species *Methanobrevibacter boviskoreani* produces methane with H_2_/CO_2_ and formate plus CO_2_ as substrates, and is susceptible to puromycin, polymyxin, and chloramphenicol [81]. Our result confirmed the susceptibility of *Methanobrevibacter boviskoreani* clade to thymol. By sequencing the genome of *Methanobrevibacter ruminantium* M1, new vaccines can be developed base on the identification of the conserved targets in methanogens [82]; the present study may suggest another promising way to suppress *Methanobrevibacter ruminantium* clade with thymol addition. The unclassified Methanomassiliicoccaceae gradually decreased and disappeared in the high dose group, indicating considerable susceptibility to thymol. As for protozoa, the proportion of *Polyplastron* and *Diploplastron-Eremoplastron* increased with a low dose and decreased with a high dose of thymol supplementation within five of the most abundant genera. In accord with the present study, Ye et al. [83] did not detect significant difference with 1000mg/d of cinnamaldehyde oil fed to Holstein cows. More specifically, our study showed proportions of *Polyplastron* differed between high dose and low dose of thymol supplemeantion which reveals the possibly different effects caused by addition levels of thymol. The genus *Diploplastron* has been reported to have high morphological similarity and phylogenetic proximity with the genus *Eremoplastron* [84]. Moreover, the typical species *Diploplastron affine* in the genus *Diploplastron* has been reported to have capacity in cellulose utilisation and digestion [85], so a high dose of thymol could affect plant mass utilisation of rumen ciliates, to some extent.

The co-occurrence network analysis revealed high correlations between rumen microbial communities and production via rumen fermentation in the current study. Degradation of carbohydrates usually produces some substrates or intermediate for methanogenesis in the rumen. In the present study, positively correlated with methane production, *Prevotella* 1 could degrade xylan, hemicellulose, and pectin; similarly, the Ruminococcaceae NK4A214 group could contribute to fibre digestion [86]. Members of the genus *Quinella* ferment carbohydrates and produce lactate, acetate, propionate, and CO_2_ as the main products [87], and thus might contribute to hydrogenotrophic methanogenesis. Interestingly, Prevotellaceae UCG-003 has been speculated to have proteolytic activity due to the function of other members in the same family; however, limited knowledge was obtained in a previous study regarding the correlation between this profile with methane production. In addition, Veillonellaceae UCG-001 has been negatively correlated to the propionate proportion [88], and a higher propionate proportion means higher efficiency in hydrogen utilisation. Consistent with the positive correlation with methane production in the present study, members of Methanomassiliicoccaceae family have been recommended as good targets for methane mitigation [89]. *Streptococcus* species in the rumen likely act as propionate-forming bacteria [90], and members of *Pseudobutyrivibrio* participate in the hydrogenation of unsaturated fatty acids [91], and thus could account for the negative correlation with methane production. No protozoa in the current study were found to be correlated with methane production, which had a positive correlation only with valerate among several VFA. Rumen function is based on cooperation between interacting rumen microorganisms. In the present study, *Treponema* 2 served as a bridge to connect other microbes that was correlated with methane production and VFA production. This finding indicates the indirect influence of rumen microbes on rumen fermentation.

## 5. Conclusions

This study investigated the responses of rumen microbes, including bacteria, archaea, and protozoa, to increasing doses of thymol supplementation, as well as parameters related to rumen function by an in vitro approach. The effective dose of thymol for methane mitigation was 200 mg/L, considering the maintenance of rumen production as well as the overall changes in rumen microorganisms. Moreover, thymol exert a stronger impact on rumen bacteria compared with archaea and protozoa seeing from the composition changes in bacteria, archaea, and protozoa. The network analysis indicates an indirect mechanism for the reduction of methane production by changing some critical bacteria members (e.g., *Prevotella* 1, *Pseudobutyrivibrio*) to supply fewer substrates for methanogenesis. The overall rumen function is based on a comprehensive interaction among rumen bacteria, archaea, and protozoa, and more work remains to be done in the future to elucidate the details of cooperation between rumen microbes.

## Figures and Tables

**Figure 1 microorganisms-08-01160-f001:**
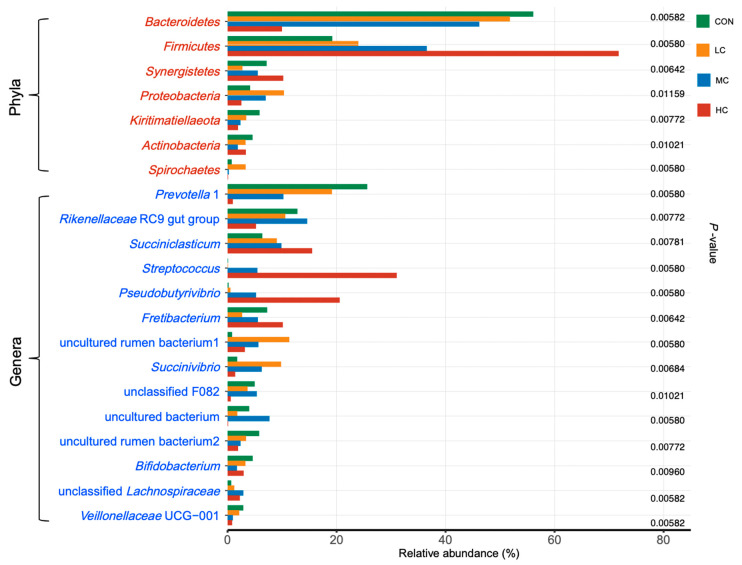
Effects of different doses of thymol supplementation on dominant bacterial taxa (average relative abundance > 1%) in vitro. Seven dominant phyla (red colored names) and fourteen dominant genera (blue colored names) are shown in the figure, the adjusted *p*-values from Kruskal–Wallis test are shown to indicate the significant difference (*p* < 0.05) with thymol supplementation.

**Figure 2 microorganisms-08-01160-f002:**
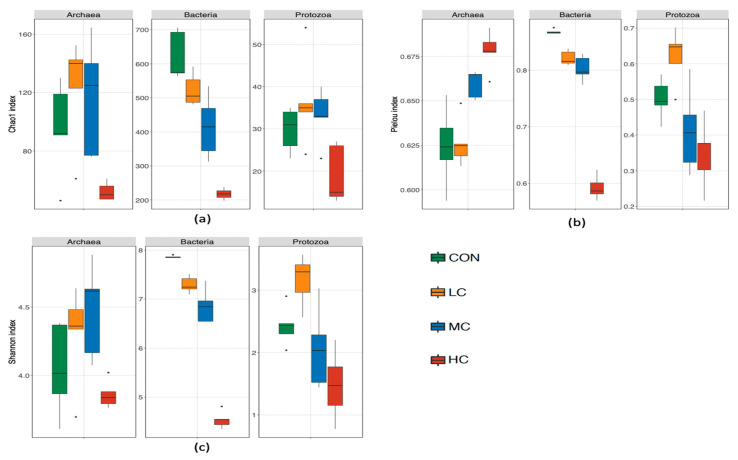
Alpha-diversity indices of rumen microbiota with different doses of thymol supplementation in vitro. (**a**) Chao1 index of archaeal, bacterial, and protozoan communities; (**b**) Pielou index of archaeal, bacterial, and protozoan communities; and (**c**) Shannon index of archaeal, bacterial, and protozoan communities.

**Figure 3 microorganisms-08-01160-f003:**
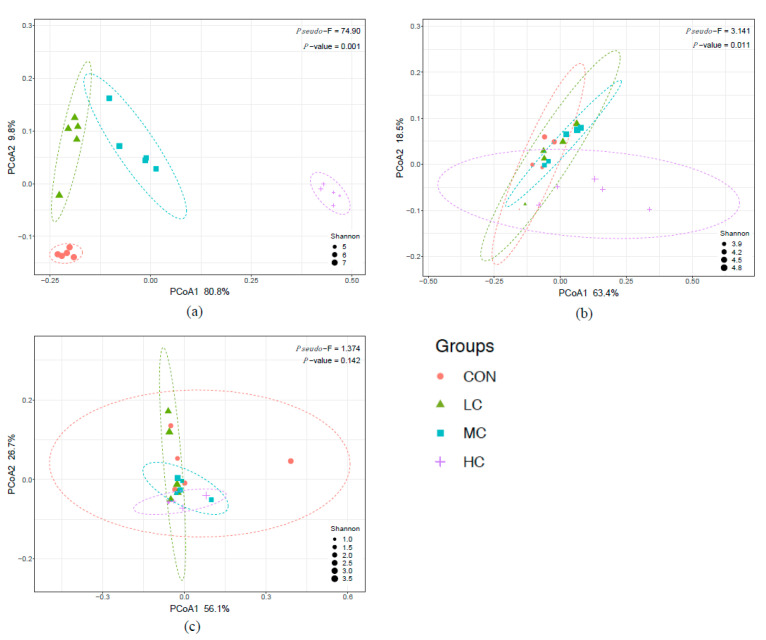
Weighted UniFrac-based Principal Coordinates Analysis (PCoA) plot of rumen bacterial (**a**)**,** archaeal (**b**), and protozoan (**c**) communities. Sizes of all the relevant samples (points) were exhibited according to values in Shannon diversity index Differences between CON, LC, MC, and HC groups were tested by PERMANOVA, significance was recognized when adjusted *p*-values were lower than 0.05.

**Figure 4 microorganisms-08-01160-f004:**
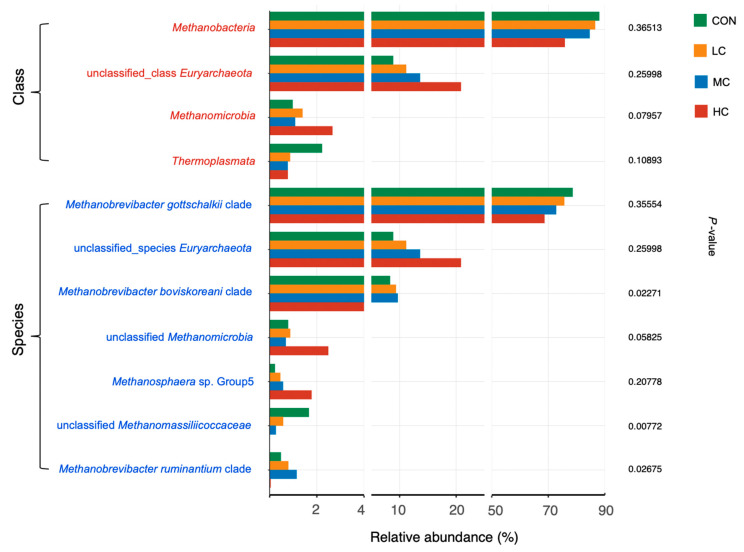
Effects of different doses of thymol supplementation on archaeal main taxa in vitro. Four dominant classes (red colored names) with relative abundance > 1% and seven dominant species (blue colored names) with relative abundance > 0.5% are shown in the figure, the adjusted *p*-values from Kruskal–Wallis test are shown to indicate the significant difference (*p* < 0.05) with thymol supplementation.

**Figure 5 microorganisms-08-01160-f005:**
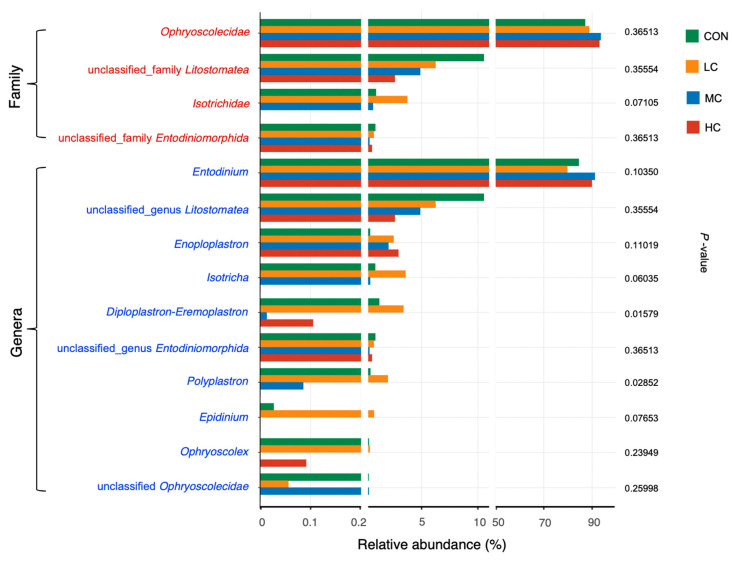
Effects of different doses of thymol supplementation on protozoa taxa in vitro. Four families (red colored names) and ten genera (blue colored names) are shown in the figure, the adjusted *p*-values from Kruskal–Wallis test are shown to indicate the significant difference (*p* < 0.05) with thymol supplementation.

**Figure 6 microorganisms-08-01160-f006:**
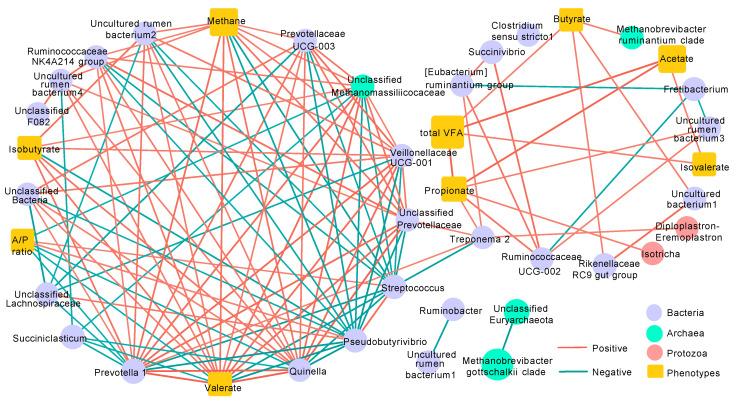
The co-occurrence network interactions of rumen bacteria, archaea, protozoa, and fermentation related parameters. The connection stands for a strong (Spearman’s ρ > 0.75) and significant (*p*-value < 0.001) correlation. The size of each node is proportional to the relative abundance of microbes as well as value of each fermentation parameter in this study.

**Table 1 microorganisms-08-01160-t001:** Oligonucleotide primers used to amplify bacterial and archaeal 16S rRNA genes, and protozoan 18S rRNA genes.

Names	Primers (Forward and Reverse) Sequence [5′→3′]	Annealing	Reference
Bacteria	338F (ACTCCTACGGGAGGCAGCA)	55 °C	[26]
	806R (GGACTACHVGGGTWTCTAAT)		
Archaea	Ar915Af (AGGAATTGGCGGGGGAGCAC)	59 °C	[27]
	Ar1386R (GCGGTGTGTGCAAGGAGC)		
Protozoa	RP841F (GAC TAG GGATTG GAG TGG)	55 °C	[27]
	Reg1302R (AAT TGC AAAGAT CTA TCCC)		

**Table 2 microorganisms-08-01160-t002:** Effects of thymol on pH, total Gas and CH_4_ production, digestibility, ammonia concentration, and volatile fatty acids (VFA) production in vitro.

Parameters	CON	LC	MC	HC	SEM	*p*-Value
pH	6.13 ^c^	6.19 ^bc^	6.23 ^ab^	6.26 ^a^	0.01	0.010
TGP (ml)	50.36 ^a^	49.22 ^a^	43.68 ^ab^	11.15 ^b^	3.72	0.006
CH_4_ (g/kg fermented DM)	53.93 ^a^	53.25 ^a^	45.56 ^ab^	11.83 ^b^	4.05	0.008
CH_4_ (g/kg fermented OM)	56.97 ^a^	54.62 ^ab^	46.62 ^bc^	12.33 ^c^	4.20	0.006
IVDMD (%)	46.31 ^a^	45.01 ^a^	43.73 ^a^	32.81 ^b^	1.48	0.020
IVOMD (%)	50.63 ^a^	50.68 ^a^	49.73 ^a^	36.34 ^b^	1.81	0.024
NH_3_-N (mg/100 mL)	14.81 ^a^	14.05 ^a^	12.57 ^ab^	9.04 ^b^	0.59	0.011
Total VFA (mmol/L)	83.80 ^ab^	93.45 ^a^	87.91 ^ab^	64.25 ^b^	3.03	0.015
Acetate (mmol/L)	34.29 ^ab^	42.36 ^a^	39.24 ^ab^	30.40 ^b^	1.47	0.032
Propionate (mmol/L)	17.09 ^ab^	19.88 ^a^	17.67 ^ab^	11.49 ^b^	0.79	0.010
Isobutyrate (mmol/L)	7.62 ^a^	6.00 ^ab^	5.83 ^b^	5.07 ^b^	0.24	0.008
Butyrate (mmol/L)	12.48 ^ab^	13.42 ^a^	14.40 ^a^	8.32 ^b^	0.66	0.012
Isovalerate (mmol/L)	5.30	5.73	5.28	4.30	0.21	0.080
Valerate (mmol/L)	7.02 ^a^	6.06 ^ab^	5.49 ^bc^	4.68 ^c^	0.20	0.006
A/P Ratio	1.98 ^b^	2.13 ^b^	2.23 ^ab^	2.65 ^a^	0.06	0.011

Note: SEM = standard error of the mean; DM = dry matter, OM = organic matter; CON = 0 thymol, LC = 100 mg/L thymol, MC = 200 mg/L thymol, HC = 400 mg/L thymol. In the same row, values with different superscript (a,b,c) indicate significant differences between treatments (*p* < 0.05), and values with superscripts containing one or two of the same letters are not significantly different.

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
