# Peer review of "Effects of Thymol Supplementation on Goat Rumen Fermentation and Rumen Microbiota In Vitro"

_microorganisms, 2020, doi:10.3390/microorganisms8081160_

Round 1
Reviewer 1 Report
In this manuscript, the authors investigated the effects of thymol on goat rumen fermentation and microbiota. I have a few comments which I would like the authors to address.
- In the method section 2.4, I would appreciate it if the authors provide more information on the which region(s) of 16S rRNA gene was targeted for PCR. E.g. V3, V4 etc.
- Could you provide references showing archaeal 16S rRNA genes could provide enough resolution for species level rather than genus level?
- The authors mentioned that “Ciliate protozoa taxonomy was assigned using navie Bayes classifier based on sequences data summarized by Kittelmann et al.” Could you provide more information on the protozoa analysis? Were the sequences data summarized by Kittelmann et al publicly available? How many sequences were included?
- Could you provide the version of Silva and RIM-DB used for the analysis?
- Did the authors use raw counts in the correlation analysis or use relative abundance of the microbiota? WGCNA was developed to find clusters of highly correlated genes. Could you provide reference showing this package is suitable for the correlation analysis of microbial taxonomy and metabolites?
- In tables, what does a, or b, c mean?
- The authors included 6 tables in the manuscript. It will be easier for the readers to understand if the authors provide more figures instead of tables. The tables could be provided as supplemental materials.
Author Response
Thanks a lot for your careful review and provide many fabulous suggestions, the point-by-point revision is attached in the word file below

Reviewer 2 Report
Major comments
In this ms authors examined gradual level of EO supplementation to goat rumen simulated culture in view of the fermentation property and microbial composition. Methodologies applied for the single experiment was rather conventional but obtained dataset and data application to an network construction are both worth receiving evaluation by subscribers. Prior to publication, however I also wonder with reluctant attitude, whether it was acceptable to allow their result presentation from one experiment with only one batch cultivation. I believe their conclusion, if they have repeated more experiment and obtained same result, could be more attractive in reliable and concrete fashion to public. I am therefore willing to see again their revision which should be supplied one more cultivation data.
Minor comments
Abstract As the first sentence it is better to describe shortly what was authors' own study purpose whereas there have been bunch of preceding data about the use of thymol into ruminant feed as well as rumen fermentation modifying agent.
L83 Need to specify the source of thymol and its chemical composition, was it a commercial product or arranged by authors?
Author Response
Thank you so much for your wonderful comments, and we provided another previous fermentation data and modified some places in the manuscript according to your comments.

Round 2
Reviewer 1 Report
My comments were addressed.
Reviewer 2 Report
I have no particular comment on their revised ms but I still found some minor English errors that need to be addressed during proofing.